# Is the Intestinal Bacterial Community in the Australian Rabbitfish *Siganus fuscescens* Influenced by Seaweed Supplementation or Geography?

**DOI:** 10.3390/microorganisms10030497

**Published:** 2022-02-23

**Authors:** Valentin Thépot, Joel Slinger, Michael A. Rimmer, Nicholas A. Paul, Alexandra H. Campbell

**Affiliations:** 1School of Science, Technology and Engineering, University of the Sunshine Coast, Maroochydore DC, QLD 4558, Australia; mrimmer@usc.edu.au (M.A.R.); npaul@usc.edu.au (N.A.P.); 2CSIRO Agriculture and Food, Bribie Island Research Centre, Woorim, QLD 4507, Australia; joel.slinger@csiro.au; 3Institute of Marine and Antarctic Studies, University of Tasmania, Launceston, TAS 7250, Australia; 4School of Health and Behavioural Sciences, University of the Sunshine Coast, Maroochydore DC, QLD 4558, Australia; acampbe1@usc.edu.au

**Keywords:** functional ingredient, immunity, core microbiome, macroalga and rabbitfish

## Abstract

We recently demonstrated that dietary supplementation with seaweed leads to dramatic improvements in immune responses in *S. fuscescens*, a candidate species for aquaculture development in Asia. Here, to assess whether the immunostimulatory effect was facilitated by changes to the gut microbiome, we investigated the effects of those same seaweed species and four commercial feed supplements currently used in aquaculture on the bacterial communities in the hindgut of the fish. Since we found no correlations between the relative abundance of any particular taxa and the fish enhanced innate immune responses, we hypothesised that *S. fuscescens* might have a core microbiome that is robust to dietary manipulation. Two recently published studies describing the bacteria within the hindgut of *S. fuscescens* provided an opportunity to test this hypothesis and to compare our samples to those from geographically distinct populations. We found that, although hindgut bacterial communities were clearly and significantly distinguishable between studies and populations, a substantial proportion (55 of 174 taxa) were consistently detected across all populations. Our data suggest that the importance of gut microbiota to animal health and the extent to which they can be influenced by dietary manipulations might be species-specific or related to an animals’ trophic level.

## 1. Introduction

Changes in the communities of microbiota within the gastro-intestinal tracts of fish (their ‘gut microbiomes’) have recently been linked to impacts on fish health and condition, including their metabolism, overall size, and immune responses [1]. This emerging understanding has great potential to facilitate the development of sustainable aquaculture industries because harnessing the positive effects of gut microbiomes on fish health could reduce the reliance of the industry on antibiotics and other chemotherapies [2]. However, we still know very little about the structure or function of gut microbiomes in most farmed fish species, or how they can be enhanced to improve yields in aquaculture [3]. Indeed, understanding the role of microbiomes in the health and resilience of marine and aquatic animals was recently highlighted as a key knowledge gap in the field of marine microbiome research [4].

Aquaculture recently replaced wild fisheries as the main source of seafood globally [5] and its importance in the provision of protein is likely to increase [5,6]. Two of the greatest threats to the sustainability of aquaculture are (1) its reliance on fish meal and fish oil from increasingly depleted wild fisheries [7]; and (2) disease [8]. The use of plant-based feed alternatives has been proposed as a potential solution to the unsustainable use of wild fish products [9], however, these novel ingredients, which fish rarely encounter naturally, can create novel challenges, such as stunted growth, increased mortality, and gut inflammation, especially in highly valuable carnivorous species [9,10].

Diet can strongly influence the structure of gut microbiomes in some fish [11,12] and dietary supplementation has been suggested as a tool to improve disease resistance in aquaculture [13]. However, dietary supplementation with plant materials from terrestrial environments can have negative outcomes, including reduced diversity in the microbial communities that persist within fish gastrointestinal (GI) tracts and associated negative health outcomes for farmed fish [13,14].

Our recent review revealed the exciting potential of seaweeds (marine ‘plants’) as immunostimulants for farmed fish, and also highlighted major gaps in our understanding, including the potential mode/s of action of successful immunostimulants (i.e., direct or mediated by the hosts’ microbiome; [15]). Indeed, the effects of most of the dietary supplements currently in use or of interest, on the microbiomes in the GI tracts of commercially important fish, are completely unknown, especially for lower-trophic level (i.e., herbivorous) fish.

We recently demonstrated that dietary supplementation with several species of seaweed caused significant stimulation of parts of the innate immune response of the mottled rabbitfish *Siganus fuscescens* [16], however, the mode/s of action of this immunostimulatory effect remain unknown. Here, we explore whether observed changes in the innate immune responses of experimental *S. fuscescens* were correlated with shifts in the GI microbiomes of those fish and thus, whether a mode of action of dietary seaweed immunostimulation may have been microbially-mediated. Seaweeds used in this trial included members of the red, green, and brown taxonomic groups and species that produce a broad range of bioactive, natural compounds (e.g., bromoform in *Asparagopsis taxiformis*, caulerpin in *Caulerpa taxifolia*; and terpenoids in *Sargassum* sp. [17,18,19]). Compounds from these species all have antimicrobial activity [19,20,21] demonstrated in laboratory assays. We hypothesised that changes in innate immune responses in *S. fuscescens* resulting from seaweed supplementation would be correlated with changes in gut microbiomes and provide evidence that a potential microbially-mediated mode of action of seaweed immunostimulants. This study targeted the hindgut bacterial community based on literature reporting that the hindgut is the part of the GI tract that contains a higher proportion of resident rather than transient microbiota [22], suggesting that this region may be more reflective of the ‘true’ fish GI microbiome.

Rabbitfish of the Siganidae family are marine herbivores presently receiving increased attention due to their attractive attributes for aquaculture [23,24,25,26] and their range-shifting ability and associated indirect impacts of warming waters on temperate ecosystems (“Tropicalization”) [27]. The mottled rabbitfish (*Siganus fuscescens*) was the focus of our study because of its candidature for aquaculture development in Asia [24] and we aimed to provide baseline information about the geographical and temporal variation of the gut microbiome of *S. fuscescens* to support the development of a sustainable farming industry for this species.

When comparing hindgut microbial communities in fish fed different diets during our experiments, we became interested in the possible existence of a core microbiome in this species. ‘Core microbiomes’ are variably defined in the literature (e.g., based on 50%, 90% or 100% prevalence cut-offs [28]) and remain a subject of active debate and research [4]. Since we know so little about natural variation in the intestinal microbiomes of farmed fish in general and herbivorous marine fish in particular, and because of our emerging interest in a possible core microbiome in this species, we also compared our data to those from two recently published studies that characterised hindgut bacterial communities from conspecific populations of *S. fuscescens*. Our aims here were two-fold: firstly, to provide some ecological context for our results and, secondly, to provide baseline data on the microbiome of Siganids to help facilitate the aquaculture development of this fish species.

## 2. Material and Methods

Mottled rabbitfish (*Siganus fuscescens*: ranging from 15 cm/70 g to 21 cm/189 g) were captured between February and March 2018 using a drag net (15 m long by 2.1 m deep with a 2.5 cm mesh size) on rocky reefs at Moffat Beach, Queensland Australia (26°47′21.7″ S 153°08′36.0″ E; Figure 1). This collection was carried out under a “General fisheries permit” (# 195305) issued by the Queensland Department of Agriculture and Fisheries (Fisheries Act 1994). Feeding trials were conducted at the Bribie Island Research Centre (BIRC) on Bribie Island, Queensland, Australia (27°03′15.9″ S 153°11′42.9″ E). After collection, fish were transferred to BIRC in an oxygenated 500 L tank. The newly captured fish were treated with hydrogen peroxide (200 mg/L for 30 min) to rid them of potential external pathogens and parasites as per BRIC biosecurity requirements. Although it was not the aim of this experiment, it is possible that this treatment might have had effects on the GI microbial communities of the fish. However, since all fish were exposed to the same hydrogen peroxide treatment, we assumed that the treatment effect was even and thus did not affect the current feeding trial. Following this, the fish were transferred to three 1000 L fibreglass tanks where they were acclimatised and fed the control (unsupplemented ‘Native’ pellets from Ridley Aquafeeds Ltd.) diet for at least two weeks. The Native diet has been formulated for Australian native carnivorous freshwater fish species and was chosen based on its low protein (38% protein, 10% fat content, and 15 MJ/kg gross energy) compared to other commercially available diets. All activities were approved by the animal ethics committee of the University of the Sunshine Coast (ANS1751).

### 2.1. Seaweed and Experimental Diets

We aimed to screen multiple species of taxonomically and chemically diverse seaweeds for their effects on the hindgut microbiomes of *S. fuscescens.* Eleven species of seaweed (5 red, 3 brown, and 3 green species) were evaluated as functional ingredients in feeding trials with *S. fuscescens* (Appendix A, hereafter referred to by genus). Four commercially available ‘aquafeed’ supplements were also evaluated: (i) Hilyses ^®^ (MarSyt Inc., Elizabethtown, PA, USA), a hydrolysed yeast culture derived from the sugarcane fermentation process (and a source of β-glucans), (ii) sodium alginate, the anionic polysaccharide extracted from brown seaweeds, (iii) the cyanobacteria spirulina (high strength organic spirulina, Swiss Wellness Pty Ltd., Collingwood, VIC, Australia) and (iv) cracked and window refractance encysted (>95%) dried biomass of the microalga *Haematococcus pluvialis*, which is a source of astaxanthin. Together there was a total of 15 supplement treatments in the trial. The proximate composition of each supplement was determined following the recommended methods of the Association of Official Analytical Chemists [30], with the protein estimation using a factor of 5 to multiply the seaweed nitrogen content as recommended by Angell, et al. [31]. The source of each species, their morphological and chemical features of interest of the supplements from the four groups (red, green, and brown seaweed and aquafeed supplements) are described in Appendix A.

For the preparation of the seaweed-supplemented diets, fresh seaweeds were rinsed with saltwater (34.5 ppt) to remove sand and biological contaminants. They were then spun in a washing machine (Fisher and Paykel 5.5 kg Quick Smart, East Tamaki, New Zealand) on spin cycle (1000 rpm) for 5 min to remove excess water, frozen at −80 °C, and then lyophilised in a freeze dryer (Thermo Savant model MODULYOD-230, Waltham, MA, USA) for 3 days at approximately −44 °C and 206 mbar. Once dried, each seaweed species was vacuum-sealed in individual bags with silica desiccant and stored at −20 °C until used. The ‘control’ (unsupplemented) diets for experimental *S. fuscescens* were produced using the commercial aquafeed ‘Native’ (Ridley Aquafeeds Ltd., Brisbane, QLD, Australia). The pellets (1.5 kg in total) were powdered then added to a blender (Hobart A120, Silverwater, NSW, Australia) with deionised water (30% weight/weight) and combined for approximately 10 min at low speed (agitator rpm of 104) using a dough hook to produce a stiff dough. The dough was extruded through a 4 mm die onto trays which were then placed in a fan-forced oven overnight at 50 °C. Once dried, the feed was packaged in airtight bags and stored at 4 °C until required. All 15 experimental diets (supplemented with seaweed or aquafeed supplement) were made in the same manner but received supplements at 3% dietary inclusion which were powdered and sieved through a 300 μm mesh prior to the addition of water during the blending step. The use of an unsupplemented, control diet is standard practice in aquaculture to test the effect that specific ingredients may have on fish [14,15,32,33,34]. The use of wild fish as control would be inadequate because wild fish are not exposed to the same conditions as fish in aquaculture (e.g., artificial diet and captivity) thus one could not conclude if differences between wild fish control and fish fed treatment diets would be a link to (1) the treatment ingredient, (2) the artificial diet used to convey that ingredient or (3) the captivity effect (e.g., exposure to artificial light and filtered seawater).

### 2.2. Experimental Design

This study used three replicate plastic tanks (55 L) for each of the 16 dietary treatments (including the control; *n* = 3) to have a total of 48 tanks, all of which were then stocked with three fish each (144 fish in total). Due to variation in sizes, fish (*N* = 144) were sorted into two size classes: ‘small’ (ranging from 15 cm/70.5 g to 18 cm/112.1 g) or ‘large’ (ranging from 18 cm/112.4 g to 21 cm/189.2 g). The 144 fish were randomly allocated into 48 tanks so that each replicate tank contained 3 fish with at least one small and one large fish. The exact mass and length of each fish were recorded and used as a covariate in analyses assessing the influence of fish diet on microbial community diversity. As processing limitations were forecasted for the end of the trial, the fish were stocked in a staggered manner with one tank per treatment stocked each day over three days to allow for the sampling of one tank per treatment each day over three days at the end of the screening trial.

Fish were fed one of 16 different diets, comprising of 15 experimental diets and a control diet. Each treatment consisted of three replicate tanks and 9 fish (3 fish per replicate tank). Therefore ‘Tank’ was a random factor nested within the fixed factor of ‘Diet’. To enable staggered sampling at the end of the experiment and ensure that all fish were exposed to the treatment diets for the same time period (two weeks), one out of three tanks from each dietary treatment was stocked with fish each day, over three days. The *Ulva* dietary treatment included only 2 replicate tanks after the loss of one tank due to water and air supply issues.

Fish were fed by hand at 3% body weight twice a day (10:00 and 15:00) for a period of 14 days. The reason for the trial lasting 14 days is based on our previous review [15], which revealed that trials where fish were fed seaweed as a functional ingredient lasted on average 14 days to conduct blood immunochemistry analysis for innate immune responses. No differences in feed consumption between tanks or treatments were observed as fish in all tanks consumed the total of both morning and afternoon feed allocations in each tank (visual inspection during handfeeding). During the trial, the water temperature was maintained at 27 °C, and the pH was within the range of 7.9 and 8.1. The system was operated as flow-through, with fresh seawater (34–35 ppt) pumped from approximately 300 m off the beach adjacent to the research station then through a series of 16 spin disk filters (40 μm) and 10 multimedia filters (~10–15 μm), after which it received ozone treatment from two 100 gO_3_/h generator units (WEDECO OCS-GSO30, Herford, Germany). The ozone-treated seawater was then pumped via ultraviolet filters, providing 80 mJ/cm^2^, to two 4 m × 2.2 m granular activated carbon vessels for a contact time of >9 min to remove unwanted by-products from the ozone treatment. Finally, the seawater was pumped to a header tank, which fed directly into a pipe system delivering treated seawater to this experiment. The system was maintained in a temperature and light controlled room kept at 24–26 °C and on a 24L:0D dim central light regime.

### 2.3. Sample Collection and Preparation

After the feeding trial (14 days), the fish were subjected to a 24 h fasting period. The fish were euthanized in 10 ppt Aqui-S^®^ (Lower Hutt, New Zealand), then the entire digestive tract from each fish was aseptically excised and placed in a Falcon tube (50 mL) before being snap-frozen and stored at −80 °C until further processing could occur.

### 2.4. Innate Immune Variables Measured

The full methodological details of samples obtained for analysis of innate immune responses to the different seaweed diets are described in Thépot, et al. [16]. Briefly, we obtained blood samples to assess cellular innate immune responses, which included the phagocytic activity/index and respiratory burst activity and we also obtained serum samples to assess humoral innate immune parameters including lysozyme activity and haemolytic activity.

### 2.5. DNA Extraction

To compare the hindgut microbiomes of fish fed different experimental diets, DNA was isolated from the hindgut and digesta of one randomly selected small and large fish from each tank (except for the *Ulva fasciata* treatment which only had 2 tanks). After the samples thawed, 0.25 g (approximately 0.5 cm length) of hindgut containing digesta was sampled. Our rationale for choosing to sample the hindgut with digesta was based on results published by Nielsen, et al. [22] and Jones, et al. [29], which suggested that this part of the microbiome was more representative of the host’s GI microbiome rather than the more transient and food associated microbiome of the midgut. We defined the section of the distal intestine starting 1 cm internally to the anal pore as “hindgut”, referred to as such hereafter. Digesta containing hindgut samples were placed directly into the isolation buffer in PowerBead tubes from the PowerSoil DNA isolation kit (Mo Bio, San Diego, CA, USA). Microbial DNA was isolated from the hindgut samples following the manufacturer’s instructions and then stored at −20 °C.

### 2.6. Sixteen S rRNA Gene Sequencing and Bioinformatics

From the isolated DNA, the 16S rRNA gene was amplified using PCR following previously published methods [35,36,37]. Briefly, the hypervariable region V3–V4 was targeted using the primers 341F (5′-CCTAYGGGRBGCASCAG-3′) and 806R (5′-GGACTACNNGGGTATCTAAT-3′) at the Australian Genome Research Facility (AGRF, Melbourne, VIC, Australia), who then sequenced the amplicons on a MiSeq platform (2 × 300 bp; MCS v3.1.0.13, San Diego, CA, USA), and the resulting reads were analysed with Illumina bcl2fastq pipeline v2.20.0.422 (San Diego, CA, USA). Demultiplexed paired-end reads were assembled by aligning the forward and reverse reads using Quantitative Insights into Microbial Ecology QIIME2 v2018.8; (available at http://qiime.org/, accessed on 10 January 2021) [38]. To ensure that comparisons were made from sequences assigned in the same hypervariable region (V4) of the comparison studies (below), the raw data from the current study was trimmed using the cutadapt package [39], using the 515F (5′-GTGCCAGCMGCCGCGGTAA-3′) and 806R (5′-GGACTACHVGGGTWTCTAAT-3′) primers as per Yu, et al. [40]. Trimmed sequences were processed and denoised using the DADA2 package v1.16.0 (available at https://www.bioconductor.org/packages/release/bioc/html/dada2.html, accessed on 10 January 2021) [41] and QIIME2 (v2018.8) software, with amplicon sequence variants (ASVs) tables constructed and aligned against the Silva 16S rRNA 99% reference database (release v132) (available at https://www.arb-silva.de/, accessed on 10 January 2021) [42]. Due to practical and budget restraints, the DNA samples were sequenced in two separate runs on the same machine at the same facility (AGRF). Bioinformatical and statistical steps were included to ensure comparability between the two sequencing runs (see below). Raw sequences have been deposited in the National Center for Biotechnology Information (NCBI) sequence read archive (SRA) under the bioproject number PRJNA649307.

Approximately 95.1% (398,112) of total reads were quality filtered and retained through this process. Subsequent quality filtering included the removal of singletons, chimeric sequences, mitochondrial DNA, and unassigned or eukaryotic ASVs. This resulted in a total of 1250 ASVs from 48 samples. Rarefaction to 6290 counts was performed to account for uneven sequencing depth among samples ( Appendix A). This resulted in the removal of one replicate from the *Laurencia* treatment (4_1_s; 874 counts) and the removal of 52 ASVs no longer present after rarefaction, leaving a total of 1,198 ASVs and 47 samples.

### 2.7. Comparisons with Previously Published Data on the Immune Response of the Same Individuals of S. fuscescens

The recently published paper [16] characterised the immune response of rabbitfish from a large experiment (*n* = 9 fish, 3 fish per tank). Here a subset of these fish per tank (*n* = 3) were randomly selected and are then related back to the immune data of those individual fish. These were included in MDS and PERMANOVA as per the Data Analysis (below).

### 2.8. Comparisons with Previously Published Sequences of the Hindgut Microbiota from Wild Populations of S. fuscescens

Two recently published papers [22,29] also characterised microbial communities in the hindgut of wild-caught *S. fuscescens*. With the permission of those authors and the provision of raw sequence data from those papers, we compared the microbiomes of our captive fish (fed experimental diets) to the results obtained from fish caught from wild populations on the east and western coastlines of Australia, respectively. Nielsen, et al. [22] characterised GI microbiomes in wild populations of *S. fuscescens* captured nearby One Tree Island (23°30′27.0″ S, 152°05′30.5″ E) in the tropical Great Barrier Reef (GBR), whereas Jones, et al. [29] sampled wild fish from two populations in Western Australia (WA), including the subtropical Shark Bay (26°01′47.28″ S, 113°33′12.49″ E) and the tropical Kimberley region (16°51′14.57″ S, 122°10′39.45″ E).

Raw sequence data were retrieved from the NCBI Short Read Archive (SRA; Jones, et al. [29]; accession number PRJNA356981 and Nielsen, et al. [22]; accession number PRJNA396430) using the SRA Toolkit software(v2.10.9, available at https://github.com/ncbi/sra-tools, accessed on 10 January 2021) and processed as demultiplexed fastq files. Raw data from both comparison studies (sequenced in the V4 region) were also processed using the cutadapt package [39] to remove respective primer sequences. From this point, the same bioinformatic pipeline as detailed above was used, with identical denoising, filtering and taxonomic reference database (Silva 16S rRNA gene 99% reference database, release v132) applied. Subsequent quality filtering included the removal of singletons, chimeric sequences, mitochondrial DNA, and unassigned or eukaryotic ASVs. Approximately 95.1% (713,284) of total reads were quality filtered and retained through this process. Subsequent quality filtering included the removal of singletons, chimeric sequences, mitochondrial DNA, and unassigned or eukaryotic ASVs. This resulted in 3160 ASVs from 86 samples. Rarefaction to 6290 counts was performed to account for uneven sequencing depth among studies and samples. This resulted in the removal of the same replicate (from the *Laurencia* treatment; (4_1_s; 874 counts) and the removal of 76 ASVs no longer present after rarefaction, leaving a total of 3084 ASVs and 85 samples (Appendix A).

### 2.9. Data Analysis and Statistics

After processing, data were imported into R v3.6.3 (available at https://www.r-project.org/, accessed on 9 October 2020) [43] using the package phyloseq [44] for statistical analysis and visualisations. The effects of the different diets and the overall relationship between 5 innate immune parameters (lysozyme activity, phagocytic activity, and index, haemolytic activity: ACH50 and respiratory burst activity), 6 health indicators (erythrocytes: RBC, leukocytes: WBC, mean corpuscular volume: MCV and hepatosomatic index: HSI), fish weight and the most abundant 17 ASVs (representing >1% relative abundance) in the hindgut of *S. fuscescens* fed the different diets was explored in a non-metric multidimensional scaling (NMDS) using Euclidian distance and compared between treatments using PERMANOVA. Differences between means are considered significant at *p* < 0.05. Alpha diversity of microbial communities, Observed ASVs, and Shannon-Weaver index (hereafter “Shannon index”), were compared among fish fed different diets and later, between different studies, using Kruskal-Wallis tests. For the rest of the analyses, to allow the comparison of both sequencing runs on a shared number of ASVs, the rarefied ASVs were agglomerated at the genus level. Venn diagrams were used to show the number of shared ASVs among samples and studies and were constructed using the Limma package [45]. Beta diversity was visualised using non-metric multidimensional scaling (NMDS) ordinations and Bray-Curtis and unweighted UniFrac community dissimilarity indices and compared between treatments and fish length as a covariate using PERMANOVA [46]. ASV level differences in relative abundance between each treatment and the control were evaluated using multiple one-way ANOVAs, with square root transformed data to meet the assumptions of homogeneity of variance and improve normality.

In order to compare our fish as one population (“Sunshine Coast”) to the other 3 wild populations of *S. fuscescens*, the 47 fish fed the different treatment diets in our trial were combined under the “Sunshine Coast” population. The four geographically distinct rabbitfish populations were analysed using pairwise comparisons of changes in the relative abundances of raw, un-rarefied data using Wald tests in the DESeq2 function [47] where the *p*-values were adjusted using the Benjamini and Hochberg method. ASV level differences between each population (Shark Bay, Kimberley, GBR, and Sunshine Coast) were evaluated using multiple one-way ANOVAs, with square root transformed data to meet the assumptions of homogeneity of variance and improve normality. Significant ANOVA (*p* < 0.05) results were followed by a Tukey’s HSD post hoc test. Additionally, the package *microbiome* [48] was used as in previous studies [49,50], to identify ASVs that were part of a core microbiome in fish from the four geographic populations. In the literature, core microbiomes are variably defined, but the most common definitions we found were that microbial taxa are considered part of a ‘core microbiome’ when they are present in 50%, 90%, or 100%, (e.g., [28]), of sampled individuals. We, therefore, applied all three prevalence thresholds to assess the possibility of a core microbiome in the hindgut of this species across studies.

## 3. Results

### 3.1. Relationship between Innate Immune Response and Microbiome Taxonomic Composition

Despite clear and often dramatic influences of seaweed diets on several innate immune parameters, there were no overall, community-level differences in the hindgut bacterial communities between fish fed the different treatment diets or any relationships between microbiota and the innate immune parameters we measured (PERMANOVA: *F* = 0.39, *p* = 0.846; Figure 2). However, 13 ASVs correlated with other measurements, including fish haemolytic activity, respiratory burst activity, and haematocrit (Figure 2B, Appendix A). The respiratory burst activity of the fish was not positively correlated with any ASV but it was negatively correlated to ASV7160 (unidentified *Firmicutes*; Figure 2A,B and Appendix A). Similarly, haemolytic activity (ACH50) was negatively correlated with ASV2081 (*Arcobacter* sp.), which was a highly abundant taxon in the hindgut of the fish fed the control diet (which also had low ACH50; Figure 3A and Appendix A). The relationship between ASV2081 and fish ACH50 is unclear as fish fed diets supplemented with *Asparagopsis* had significantly higher ACH50 than the other fish [16], although there were no differences in the relative abundance of this ASV between diet types. Rather, seven treatments had higher relative abundance while eight treatments had a lower relative abundance of ASV2081 compared to the fish fed *Asparagopsis* (Appendix A).

### 3.2. Bacterial Community Diversity

In total, we recovered 1198 ASVs after rarefaction from the hindgut of *Siganus fuscescens* (*N* = 47) used in our experiment. To allow the comparison of both sequencing runs from this trial at the ASV level, the rarefied ASVs abundance agglomerated at the genus level. This left 113 assigned genera in total from all 5 treatment groups (red, green, brown seaweed, and aquafeed supplements and control). Out of these 113 taxa, the hindgut of the fish fed the control and supplemented diets shared 63 taxa (Figure 4), with further overlaps with and between the seaweed groups (Figure 4). Hindguts of fish fed control diets had fewer taxa (69 taxa) compared to fish fed supplemented diets, which were all similar with 98 taxa in fish fed red seaweeds, 95 taxa in fish fed green seaweeds, 93 taxa in fish fed ‘aquafeed’ supplements and 87 taxa in fish fed brown seaweeds (Figure 4).

Although there appear to have been some dissimilarities, there were no statistically significant differences in alpha or beta diversity indices between treatments (*p* > 0.05; Figure 5A,B) or between treatment groups (*p* > 0.05; Figure 5C,D).

The beta diversity results (PERMANOVA based on Bray-Curtis and unweighted UniFrac measures; *p* = 0.99 and *p* = 0.209 respectively) did not show clear differences between the composition of hindgut microbial communities in fish fed different diets compared to the control fish (Figure 6). There was also no effect of fish size (weight or length) on the microbiome when each size variable was added (independently) as a covariate in separate PERMANOVAs (weight: *p* = 0.104 and length: *p* = 0.15 for unweighted UniFrac measure).

### 3.3. Microbiome Taxonomic Composition

Of the 113 taxa detected post rarefaction and agglomeration to the genus level, only 17 represented more than 1% of the total abundance. Of those 17 taxa, eight belonged to the phylum *Firmicutes*, three belonged to the *Proteobacteria*, and another two each to *Bacteroidetes* and *Fusobacteria*. The most abundant phylum (*Proteobacteria*) was represented by just three taxa and accounted for an average 44.0% ± 1.42% relative abundance across our samples (Mean ± SE). The most abundant taxon was the genus *Desulfovibrio* and represented between 11% (sodium alginate fed fish) and 34% (*Dictyota* fed fish; Figure 3A). The second most abundant phylum was the *Firmicutes* with 20.4% ± 1.4% abundance. *Firmicutes* was the only phylum that differed significantly between the different diets in our screening trial (ANOVA, *F* = 3.07, *p* = 0.003), with the lowest relative abundance observed in the hindgut of fish fed the control diet (9.3% ± 4.0%) compared to an average of 21.1% ± 1.4% for all other treatments (Figure 3A). Fish fed the *Haematococcus* sp. and *Halimeda* sp. diets had the highest relative abundance (28.8% ± 4.2% and 28.5% ± 9.9% respectively) of *Firmicutes* and the average value for the seaweed supplements was 20.5% ± 1.7%. Conversely, fish on the control diet seemed to possess a higher proportion of bacteria in their hindgut belonging to the phylum *Epsilonbacteraeota* (Figure 3A) although this consistent observation could not be resolved statistically.

At the family level, there were no significant differences between fish from the different diet treatments except for the relative abundance of *Ruminococcaceae* (3.5% ± 1.6%; *F* = 3.08, *p* = 0.004) which was lower in fish fed control diets compared to those fed supplemented diets, with the highest relative abundance for that taxon observed in fish fed the calcified green seaweed *Halimeda* sp. (13.0% ± 6.3%; ANOVA, *F* = 3.07, *p* = 0.003). The relative abundance of bacteria from the *Arcobacteraceae* family also appeared to be higher in hindguts of fish fed control diets (Figure 3A and Appendix A). However, despite the magnitude of these differences, they were not significant when compared to supplemented diets overall or individually.

Although, most of the bacterial genera in our samples were unidentified (53%), we did find some differences between the communities in the hindgut of fish fed any of the supplemented diets compared to those fed the control diet at the genus level. For example, although the relative abundance of *Fusobacterium* spp. was low and variable across all fish, including those fed with supplemented diets (average 1.6% ± 0.5%), it had extremely low abundance (0.3% ± 0.2%) in the hindgut of fish fed the control diet (ANOVA, *F* = 2.135, *p* = 0.036; Appendix A).

### 3.4. Comparison with Wild Populations

To provide ecological context for our results and additional information for the development of bespoke aquafeed for this fish species, we compared our microbiome data, obtained from fish collected on the subtropical Sunshine Coast, Australia, in 2018, to those obtained from conspecific hindgut microbiomes from individuals in populations located ~4000 km west off the Western Australian (WA) coastline (Shark Bay and the Kimberley Coast) and ~350 km north on the Great Barrier Reef (GBR; One Tree Island) during 2015 and 2016. In total, we recovered 3084 ASVs after rarefaction (6290 sequence depth; Appendix A) from hindguts of *S. fuscescens* from the three combined studies (85 samples in total; with *n* = 47 fish from our study, *n* = 16 from Nielsen, et al. [22] and *n* = 22 (made of fish from Kimberley *n* = 16, and from Shark Bay *n* = 6) from Jones, et al. [29].

To compare the three studies, taxa agglomeration was performed at the genus level. This led to a comparable list of 174 taxa in total, including 134 taxa that were present in our study on the Sunshine Coast, 110 taxa that were detected in fish from the GBR; Nielsen, et al. [22], 110 taxa that were detected in Shark Bay and 130 present in the Kimberley (WA) fish; Jones, et al. [29] (Figure 7). We found 55 ASVs that were common in hindguts of mottled rabbitfish from all three populations and 35 ASVs that were present in 50% of all samples (Figure 7 and Appendix A). When the prevalence threshold was increased, only 9 ASVs were found to be shared between 90% of the individual fish and only 6 ASVs were present in 100% of individuals sampled across all three studies (Appendix A). Fish from the Sunshine Coast had the highest number of unique taxa (i.e., those that we did not detect from other populations) followed by those from the GBR (Figure 7). Fish from the two populations in WA shared more taxa than any other two *S. fuscescens* population. Fish from the Sunshine Coast shared more taxa with the GBR fish than either of the two WA populations (Figure 7). There was a marginally non-significant difference in alpha diversity between the four populations in terms of the number of observed ASVs: *F* = 7.30, *p* = 0.06) and the Shannon index: *F* = 85.81, *p* = 0.12) of *S. fuscescens* (Figure 8).

The analyses of beta diversity revealed strong differences between the four populations of *S. fuscescens* (PERMANOVAs based on Bray-Curtis and unweighted UniFrac measures; *F* = 13.39, *p* = 0.001 and *F* = 16.34, *p* = 0.001 respectively; Figure 9). Omitting the data from our trial (Sunshine Coast), the hindgut microbiome of the other three *S. fuscescens* populations (Shark Bay, Kimberley, and GBR) were also significantly different from each other (PERMANOVAs, *F* = 14.87, *p* = 0.001, and *F* = 8.97, *p* = 0.001 for Bray-Curtis and unweighted UniFrac respectively; Figure 9A,B). The beta diversity of the two wild populations from Western Australia (Shark Bay and Kimberley) differed significantly based on Bray-Curtis (PERMANOVAs, *F* = 2.66, *p* = 0.009) but not on UniFrac (PERMANOVAs, *F* = 1.14, *p* = 0.29).

### 3.5. Microbiome Taxonomic Composition of the Three Studies

The three studies were clearly distinguishable from each other with respect to the relative abundance of many differentially abundant ASVs (Figure 3, Appendix A and Appendix A). Fish from the GBR appeared to have the highest relative abundance of *Proteobacteria* (52.6% ± 2.9%) compared to all the other fish including those in our study (30.9% ± 2.2%) and the fish from WA (26.5% ± 1.9% and 20.6 % ± 2.0% for Shark Bay and the Kimberley fish respectively), however, this was not resolved statistically (ANOVA, *F* = 1.45, *p* = 0.236; Figure 3B). The GBR fish were the only ones without ASVs from the *Spirochaetes* (ANOVA, *F* = 69.08, *p* < 0.001), and they also had the lowest relative abundance of *Fusobacteria* (0.2% ± 0.0%, ANOVA, *F* = 6.86, *p* < 0.001) and *Epsilonbacteraeota* (0.06% ± 0.02%, ANOVA, *F* = 3.34, *p* = 0.023), which across all the other fish represented an average of 3.8% relative abundance (Figure 3B). The four most abundant ASVs in the GBR fish represented 75.2% ± 2.2% of the relative abundance compared to 71.9% ± 2.2% and 61.3% ± 1.9% in Western Australia and the Sunshine Coast fish, respectively (Figure 3B and Appendix A).

Fish from Western Australia had more similar hindgut microbiomes to the fish from our feeding trial compared to those from the GBR. For example, the relative abundance of *Fusobacteria* in Shark Bay fish was comparable to those in our study, with (13.6% ± 3.5% and 10.6 % ± 0.7% respectively, Tukey ‘Sunshine Coast vs Shark Bay’ adjusted *p* = 0.778) but it was significantly lower in those from the Kimberley (9.6% ± 1.2%, Tukey ‘Shark Bay vs. Kimberley’ adjusted *p* = 0.049) and the GBR (6.1% ± 0.6%, Tukey ‘Shark Bay vs. GBR adjusted *p* = 0.018; Figure 3B and Figure 8). Furthermore, the *Bacteroidetes*, which represented 15.0% ± 2.9% of the community in the hindgut of fish from the Kimberley and 16.2% ± 0.8% from fish in our study (Tukey ‘Sunshine Coast vs Kimberley’ adjusted *p* = 0.757), significantly higher than for GBR fish (Tukey ‘Sunshine Coast vs GBR’ adjusted *p* = 0.0.24).

One key difference between our study and the others was the absence or very low relative abundance of *Spirochaetes* (0–0.17% ± 0.0%; ANOVA, *F* = 69.08, *p* < 0.001) in the GBR and WA fish, compared to our samples which had a relative abundance of 5.7% ± 0.5% (Tukey ‘Sunshine Coast vs GBR’, ‘Sunshine Coast vs Shark Bay’, ‘Sunshine Coast vs Kimberley’ adjusted *p* < 0.001; Figure 3B). Furthermore, 65 ASVs significantly differed (Wald tests, adjusted *p* < 0.05) in abundance between the hindgut microbiome of one or more of the four geographically distinct populations of rabbitfish from the 3 studies (Appendix A and Appendix A). Compared to the GBR and WA, our fish tended to have increased relative abundances of ASVs representing >1% abundance with assigned genera. These ASVs included *Treponema* spp., *Romboutsia* spp., *Turicibacter* spp., and *Ruminococcaceae* UCG-014, which all consistently and significantly represented greater proportions of the microbial communities in the hindguts of our fish compared to the other populations (Appendix A and Appendix A).

There were some exceptions to this pattern. For example, abundances of *Akkermansia* spp. and *Tyzzerella* spp. were significantly lower in our fish than in the other studies (ANOVA, *F* = 69.08, *p* < 0.001; Appendix A and Appendix A). The fish from both sites in WA also had significantly higher relative abundances of *Rikenella* spp. and *Sedimentibacter* spp. than our fish, while the GBR population had higher relative abundances of *Terrisporobacter* spp. and *Staphylococcus* spp. than any of the other geographical locations (Appendix A and Appendix A). The most similar populations were those from the two sites in WA (only 12 significantly different ASVs between those; Appendix A). Our fish appeared to be most similar to those from Shark Bay in WA, which had only 27 ASVs with significantly different relative abundances compared to the other two populations which had more than 53 (Appendix A and Appendix A). The two populations with the highest amount of ASVs which significantly differed between their hindgut were the fish from the GBR and those from our study with 65 different ASVs (Appendix A and Appendix A).

Despite those differences, the core microbiome analyses revealed that out of the 55 shared assigned taxa between the three studies, 35 were present in 50% of all the fish sampled (43 out 86 fish; Appendix A). Only seven out of the 35 assigned taxa could be assigned to genera and these included *Fusobacterium* sp., *Romboutsia* sp., *Treponema* sp., *Arcobacter* sp., *Alistipes* sp., *Odoribacter* sp., and *Brenakia* sp. (Appendix A). Finally, 13 taxa represented between 66% and 85% of the total relative abundance in the hindgut of *S. fuscescens* regardless of its geographical population. When the prevalence threshold was increased to 90%, *Alistipes* sp. was the only taxon out of 9 ASVs that were identified at the genus level, while the six taxa that were present in all individual fish were unidentified at the genus level (Appendix A).

## 4. Discussion

Supplementation of diets with seaweeds and commercially available aquafeed supplements had only subtle effects on the diversity and composition of hindgut microbial communities in the rabbitfish *Siganus fuscescens*. Therefore, none of the strong, seaweed-derived immune responses reported previously [16] showed any correlations to changes in microbiome diversity or composition. The only exception was the bacterial genus *Fusobacterium*, which was enhanced in the hindgut of fish fed diets supplemented with seaweed or other functional ingredients. This result is surprising and diverges from an emerging understanding of links between gut microbiomes and health and immunity in animals. In our study, hindgut microbiomes remained remarkably consistent between treatments including the control fish suggesting that (i) the dietary supplementations (3% dietary inclusion) which led to profound immune responses in the fish were insufficient to elicit a strong change in the fish GI microbiomes, (ii) the immune response we observed in experimental fish was unlikely to have been microbially mediated, (iii) there is potential for the existence of a stable, core GI microbiome in *S. fuscescens*, for which we conducted a further investigation with published data from two other studies [22,29].

Despite some clear and expected differences between the GI microbiomes in fish from our study and the two other studies, 55 out of 174 assigned taxa were shared, and 13 of those represented between 66% and 85% of the total relative abundance in all fish. These observations from geographically and temporally distinct populations—including our fish which were all fed experimental diets based upon commercial fish pellets developed for carnivorous fish (with and without seaweed and other supplements)—provide compelling, initial evidence for a possible core microbiome in this opportunistic omnivorous subtropical fish species. The hindgut microbiome of *S. fuscescens* appears surprisingly stable, despite experimental manipulations of diet at levels that are known to be able to fundamentally change the outcomes of production and other fish traits [13,51].

### 4.1. Lack of Correlation between Microbiomes and Innate Immune Responses

The number of studies exploring the effect of seaweed dietary supplements on both the immune and gut microbiome of fish is limited [15]. However, there is evidence of shifts in bacterial communities associated with fish gastrointestinal tracts after seaweed treatment (Thepot et al., 2021b). These shifts tended to be associated with improved immune responses, including up-regulation of immune-related genes [32,52]. Several studies also reported that fish fed seaweed had reduced levels of potentially pathogenic bacteria including *Aeromonas hydrophila* [53,54] and improvements in humoral immune defences including lysozyme activity and respiratory burst activity [54]. However, in our trial, those seaweed species that induced strong immune responses in *S. fuscescens* (e.g., *Asparagopsis taxiformis* and *Dictyota intermedia* [16]) were not correlated to any changes in hindgut microbial composition from the same experiment.

These observations suggest that the effects of seaweeds, especially *Asparagopsis taxiformis* on the immune responses in *S. fuscescens*, were direct and not mediated by microbiomes in the hindgut of the fish. However, it is possible that the seaweed dietary supplements had effects on the microbiomes of the fish outside of their hindgut, including the skin and gills, which have previously been reported to be locations where the microbial community can be influenced by diet [55]. Furthermore, we preferred to measure in situ immune responses through various immunochemistry tests rather than their gene expression as a proxy for immune stimulation because the relationship between mRNA transcripts and protein abundance is often quite low (~30–40%) [56]. It is possible that the immune related genes of the fish fed the seaweed diets might have correlated with the observed changes in the fish hindgut microbial communities as per previous studies [32,52]. Future studies investigating the mechanisms involved in fish immunostimulation should include measurements of both the fish immunochemistry, their GI microbiome, and the relevant immune-related gene expression.

### 4.2. Effects of Diet on the Hindgut Microbiome of S. fuscescens

The subtle effects of the dietary supplements in this trial are surprising considering the potent immunostimulatory effects some of the supplements had and their diverse natural product composition [16]. Some previous studies have observed dramatic effects of experimental diets on the intestinal microbiome of farmed fish with comparable inclusion rates and experimental designs (i.e., *Sparus aurata* and *Seriola lalandi* [57,58,59]), whereas others found that the hindgut microbiome of cultured fish was relatively stable and did not appear to show much overall change to dietary manipulation (i.e., *Oncorhynchus mykiss* and *Siganus canaliculatus* [26,33,60]). The studies that detected strong changes typically supplemented fish diets with probiotics or other functional feeds for 4–8-week experiments, longer than our experiments but with a similar dietary inclusion ratio. On the other hand, Wong, et al. [60] fed rainbow trout experimental diets including grains for a period of 10 months and observed only subtle changes in the fish intestinal microbial communities. Lyons, et al. [33] supplemented the diet of rainbow trout for 15 weeks but in this case with a microalgal meal at a level of 5% and found that whilst addition increased diversity, the overall structure of microbiomes in the hindgut of the fish were not significantly altered. Similarly, Zhang, et al. [26] found that supplementing *Siganus canaliculatus* (a color morph of *S. fuscescens* [61]) with 10% of the green seaweed *Ulva pertusa* for a period of 8 weeks did not significantly alter the microbial diversity in the intestinal communities in the fish. They concluded that a strong core microbiome existed, comprised of 86 operational taxonomic units, which were shared across their fish regardless of the dietary treatment [26]. Out of the 86 OTUs reported in Zhang, et al. [26], three, namely *Arcobacter* sp., *Fusobacterium* sp., and *Treponema* sp. were also identified as core members of *S. fuscescens* hindgut microbiome.

Although seaweed is a highly diverse group (>10,000 species) that produce a wide range of secondary metabolites with bioactive properties and are increasingly used as animal supplements, there is a gap in the literature regarding their potential as a dietary supplement to shape the intestinal microbiome of animals including fish [62,63]. Ours was the first study to test so many seaweed species in one experiment, including several known for their bioactive natural products (e.g., *Asparagopsis taxiformis*, *Caulerpa taxifolia,* and *Sargassum* spp.). Indeed, there was some evidence for the effects of these chemically ‘rich’ seaweed supplements. For example, fish fed diets supplemented with the green alga *Caulerpa taxifolia* had significantly higher levels of *Fusobacterium* spp. and similarly enhanced *Cetobacterium* spp. and *Treponema* spp. *C. taxifolia* produces many interesting bioactive compounds [21] and its presence on reefs can completely alter sediment microbiomes through chemical modifications of the substrate (see [64] and references therein). This seaweed is typically avoided by native herbivorous fish (e.g., *Girella tricuspidata*) and invertebrate grazers in Australia [65] and can be toxic to invertebrates forced to consume it in feeding trials [65,66]. The red seaweed *A. taxiformis* was also expected to significantly modulate the GI microbiome of the rabbitfish fed that supplement due to its production and storage of potent antimicrobial bioactives [20] and its fast modulatory effect on the rumen microbes of ruminants [67,68]. However, the abundance of some ASVs was only slightly enhanced in fish from that treatment (e.g., *Romboutsia* sp.) and these changes were not statistically significant.

### 4.3. Comparing Separate Studies of the Hindgut Microbiome of S. fuscescens

Not surprisingly, the geographically distinct populations of *S. fuscescens* that were sampled by different teams at different times from different places had significantly different hindgut microbiomes. However, there was still substantial overlap, with almost 18% of bacterial taxa present across all populations. Interestingly the hindgut microbiome of our fish appeared to have more in common with that of the fish from Western Australia (~4000 km away) than with the geographically closer population of *S. fuscescens* from the GBR (~380 km away), which seemed more distinct than other populations. Overall, the hindgut microbiome composition of the fish from our study (Sunshine Coast) was most similar to those from Shark Bay in Western Australia, which is at a similar latitude.

Potential explanations for these groupings are that all of the seaweed genera fed to our fish have tropical, subtropical, or temperate distributions and are common on the eastern coast of Australia, with many also occurring on the west [69]. It is, therefore, possible that some of the similarities between populations were the result of similar native diets that include these seaweed species. Another explanation for the similarities observed between our fish, and those from Shark Bay more specifically, could be the similar abiotic and biotic factors, given that the Kimberley and GBR sites are both tropical and the Sunshine Coast and Shark Bay sites are both sub-tropical locations. Furthermore, the fish from our screening trial were collected near shore (<1 km away), as were the fish from Shark Bay, whereas the fish from the Kimberley site was approximately 25 km from the coast and finally the most distinct hindgut microbiome was found in the fish from One Tree Island on the GBR which is about 70 km from the coast. The impact of rivers, agriculture, and other human or land-associated impacts may be more important in nearshore areas, which could explain some of the differences observed here. This hypothesis is further supported by the fact that the diet of the fish in our trial (97% commercial pellet designed for carnivorous fish with 3% seaweed inclusion) would be drastically different from that of the wild fish populations which predominantly would feed on seaweed [22,29]. Similarly, low spatial and temporal variation was observed in the gut microbiomes of larvae from another rabbitfish species, *Siganus guttatus*, across three sites separated by up to 390 km across a three-year sampling program [70]. This observation of stability in the microbiome led the authors to propose that *Siganus* spp. have a core microbiome in their GI tracts [22,26,34,70].

### 4.4. Does Siganus Fuscescens Have a Core Microbiome?

Despite experimental manipulations of diets with taxonomically and chemically diverse seaweeds and samples originating from populations in locations separated by up to 4000 km around the Australian coast (the Sunshine Coast in southeast Queensland; present study, Shark Bay and the Kimberley site in Western Australia; Jones, et al. [29], and the Great Barrier Reef in North East Queensland; Nielsen, et al. [22]), more than 50% of all mottled rabbitfish included shared nearly one-third of their hindgut microbiota, with between 6 and 35 taxa belonging to a potential core microbiome in this species, depending upon which threshold is used [28]. Definitions of a ‘core microbiome’ are still the topic of debate and disagreement in the literature [71], however, the identification of a core microbiome was recently highlighted as one of the first steps required to link microbial community structure and diversity to its function and, importantly, the role it plays for its host [4]. Our observations provide further evidence for the potential existence of a core microbiome in this species and support previous suggestions that *Siganus* species may have a core microbiome that is robust to dietary manipulations and large geographical distances as per other *Siganus* spp. [22,26,34,70]. By helping fish maintain homeostasis in new and changing environments, the existence of a core microbiome could confer performance advantages to fish in aquaculture settings and could also be a mechanism for their success as tropical invaders into temperate waters. However, further, more targeted work is needed to confirm whether *S. fuscescens* does indeed have a core microbiome and importantly, the functional roles of any core microbial taxa throughout the life of *Siganus* fish in the wild and on farms.

## 5. Conclusions

Immunostimulatory effects of dietary supplementation with seaweeds in the mottled rabbitfish *Siganus fuscescens* appear not to be microbially mediated. Rather, fish had remarkably stable hindgut microbiomes that were only subtly influenced by dietary manipulation with diverse seaweeds (including several with highly bioactive natural products) and commercial products. These results are contrary to emerging studies from other fish species and animals-including humans-and suggest that the effects of diet and functional feeds on animal gut microbiomes and resulting health may be species-specific and influenced by trophic levels. Our observations provide some preliminary evidence that a conserved core microbiome may exist within the hindgut of this fish species and provide other baseline data about temporal and spatial variation in the hindgut bacterial communities within this candidate aquaculture species, which may support the sustainable development of this industry.

## Figures and Tables

**Figure 1 microorganisms-10-00497-f001:**
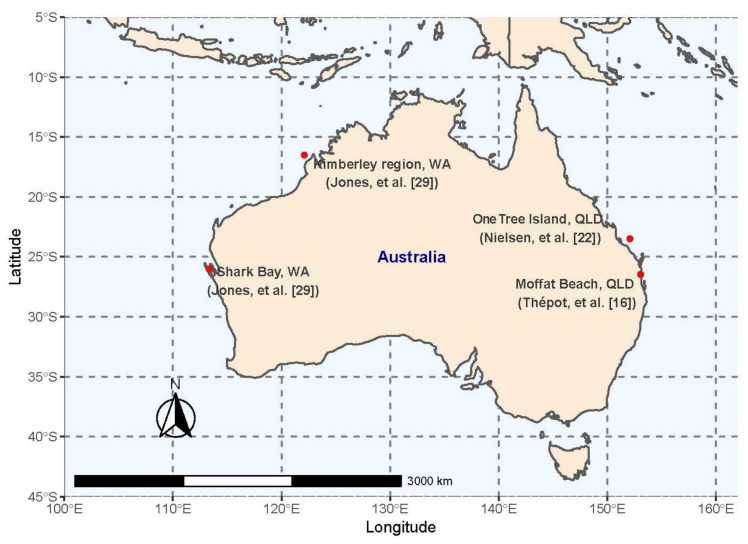
Map of mainland Australia including the four *Siganus fuscescens* sampling locations from the three studies; Kimberley region (pink) and Shark Bay (dark pink) from Jones, et al. [29], One Tree Island (blue) from Nielsen, et al. [22] and Moffat Beach (black) from Thépot, et al. [16].

**Figure 2 microorganisms-10-00497-f002:**
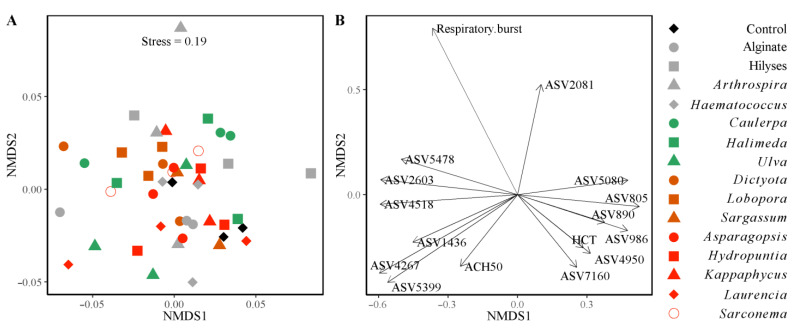
(**A**) Non-metric multidimensional scaling (NMDS) plot (Euclidean distance) of the 5 innate immune parameters (lysozyme activity, phagocytic activity and index, haemolytic activity: ACH50 and respiratory burst activity), the 6 health indicators (erythrocytes: RBC, leukocytes: WBC, mean corpuscular volume: MCV and hepatosomatic index: HSI), fish weight and most abundant 17 ASVs (representing >1% relative abundance) in the hindgut of *S. fuscescens* fed the different dietary treatments with the individual fish and (**B**) plot of the original variables (innate immune response and health indicators) loaded as vectors in NMDS space (with loading >0.7; *p* < 0.05). The different colours represent the fish fed the different groups of seaweed (green, brown, and red), the positive controls (grey), and those fed the unsupplemented control diet (black) (*n* = 3 data points per treatment refer to 3 replicate tanks comprised of one sub-sampled fish [small or large]).

**Figure 3 microorganisms-10-00497-f003:**
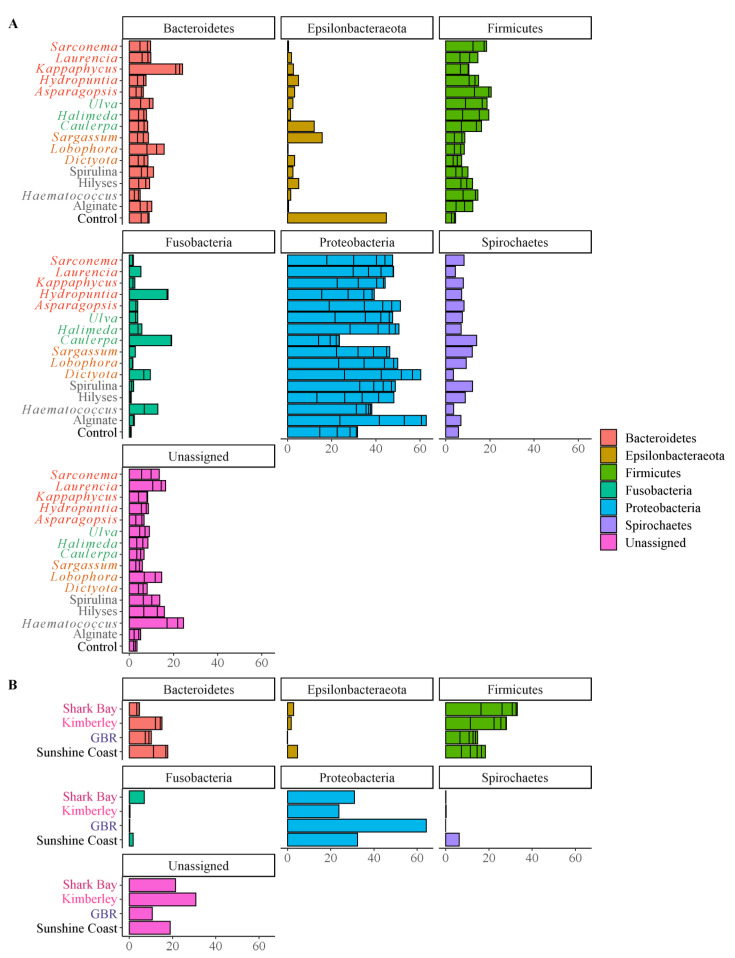
Mean relative abundance of phyla in the hindgut of the fish fed the different treatments and from the different geographical populations of *S. fuscescens*. Phyla contributing to >1% abundance to the microbial communities of the hindgut of *S. fuscescens* (**A**) fed the 16 diets of the current study and (**B**) of the fish from the current study and those from the three wild populations of this fish. On y-axes, red text indicates that diets were supplemented with a species of red seaweed, green text indicates green seaweed, and brown text indicates supplementation with brown seaweed. Aquafeed supplements are indicated in light grey with the control in black. The fish from Eastern Australia (GBR) are in blue and those from Western Australia (Shark Bay and Kimberley) are in pink.

**Figure 4 microorganisms-10-00497-f004:**
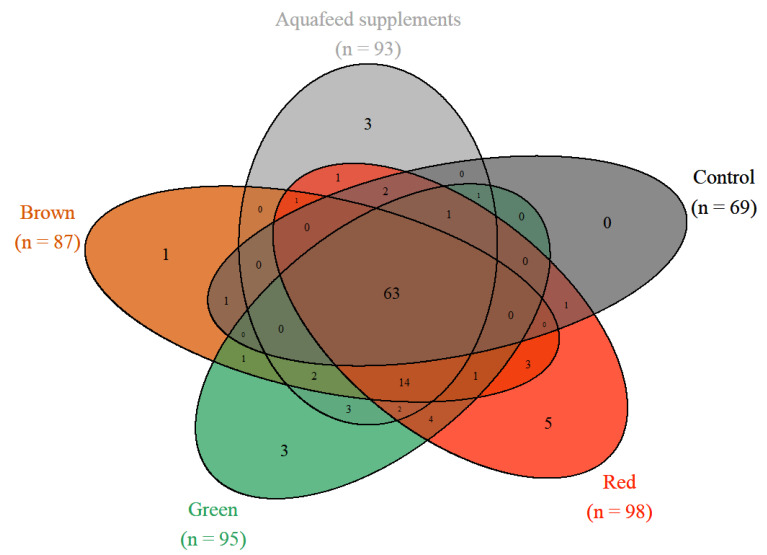
Venn diagram of the shared ASVs between the hindgut of the fish fed the different diets based on their functional groups. Shared and unique ASVs in the hindgut of the mottled rabbitfish (*S. fuscescens*) fed the control diet or diets supplemented with reds (*N* = 15), greens (*N* = 9) or browns (*N* = 9) seaweeds or aquafeed supplements (existing industry dietary additives; *N* = 12).

**Figure 5 microorganisms-10-00497-f005:**
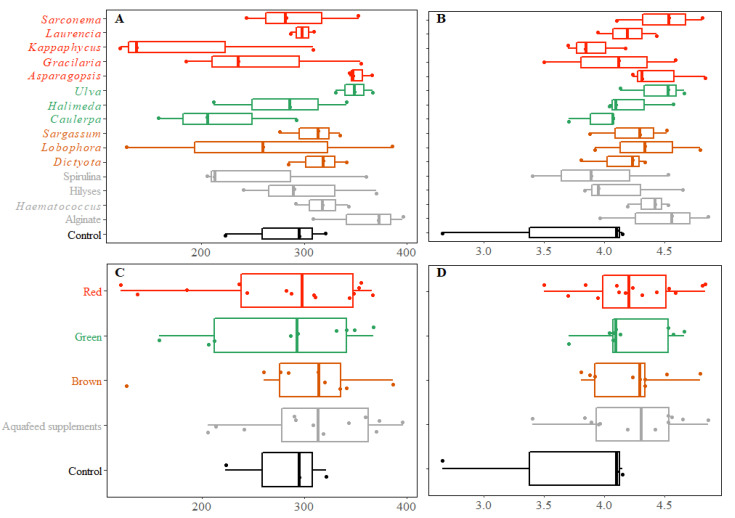
Alpha diversity indices in relation to dietary treatments. Alpha diversity analysis using species richness (Observed ASVs; (**A**,**C**)) and species diversity (Shannon index; (**B**,**D**)) for all treatments (**A**,**B**) and for the different functional groups of seaweeds used in feeding trials (**C**,**D**).

**Figure 6 microorganisms-10-00497-f006:**
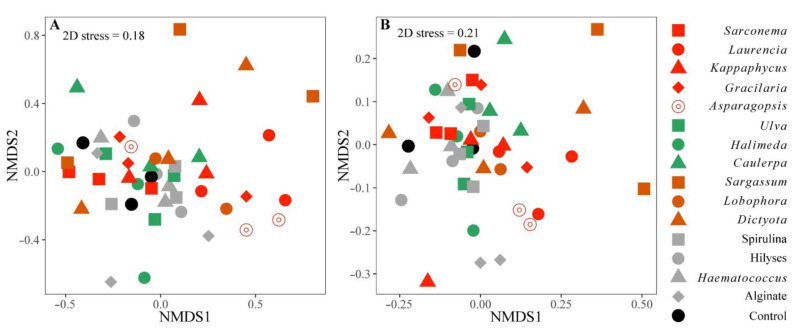
Beta diversity indices in relation to the dietary treatments using NMDS on rarefied ASVs abundance using Bray-Curtis (**A**) and unweighted UniFrac (**B**) dissimilarities between the genus-subset hindgut bacterial communities of *S. fuscescens* fed the supplemented or control diets. Symbol colours correspond to diet treatment type, including brown seaweed (brown symbols), red seaweed (red symbols), green seaweed (green symbols), Aquafeed supplements (grey symbols), and control diets (black symbols).

**Figure 7 microorganisms-10-00497-f007:**
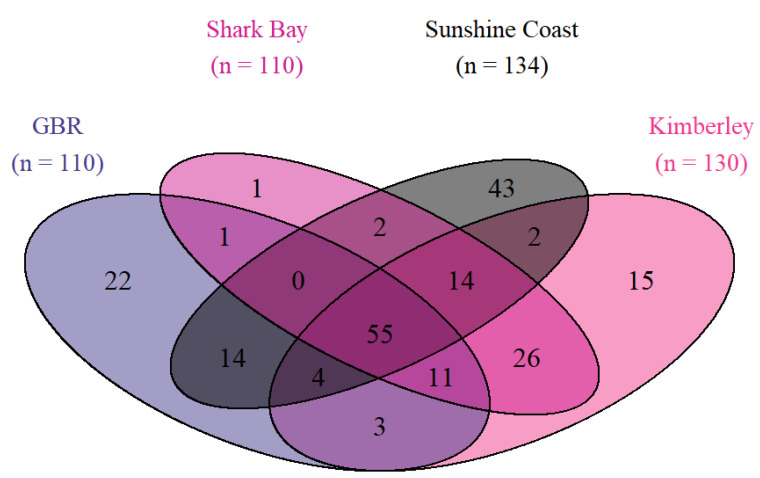
Venn diagram of the shared ASVs between the hindgut of the fish from the different geographical populations. Shared and unique ASVs in the hindgut of the mottled rabbitfish (*Siganus fuscescens*) from the current study (Sunshine Coast: *N* = 47) and the three wild *S. fuscescens* populations (GBR: *N* = 16; Kimberley: *N* = 16 and Shark Bay: *N* = 6).

**Figure 8 microorganisms-10-00497-f008:**
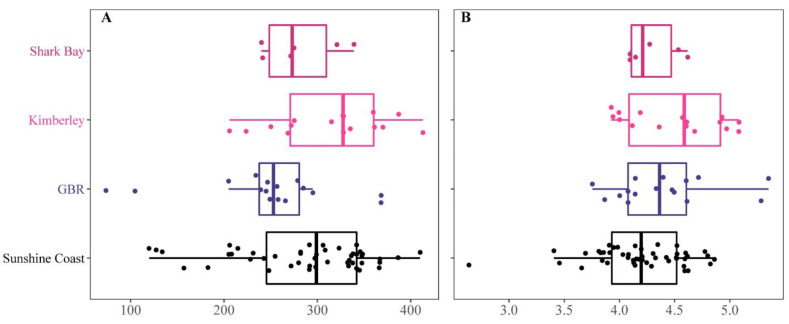
Alpha diversity indices in relation to the fish’s geographical population. Alpha diversity analysis using species richness. Observed ASVs; (**A**) and species diversity. Shannon index; (**B**) from the four *Siganus fuscescens* populations.

**Figure 9 microorganisms-10-00497-f009:**
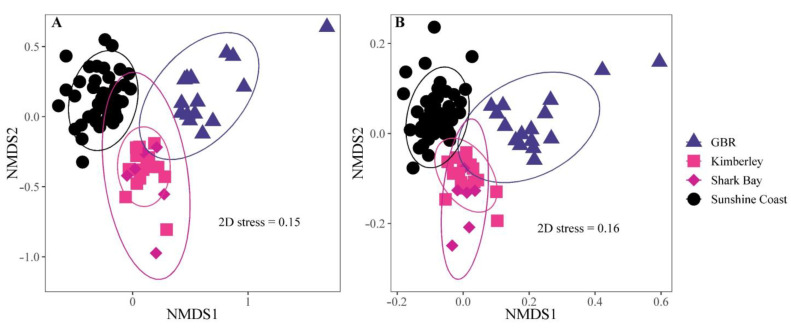
Beta diversity indices in relation to the fish’s geographical population. NMDS based on Bray-Curtis (**A**) and unweighted UniFrac (**B**) similarities of the rarefied ASV abundance in the hindgut of *S. fuscescens* individuals collected by Nielsen et al., (GBR) Jones et al., (Kimberley and Shark Bay) and the current study (Sunshine Coast). The ellipses represent the 95% confidence interval.

## Data Availability

Raw sequences have been deposited in the National Center for Biotechnology Information (NCBI) sequence read archive (SRA) under the bioproject number PRJNA649307. Raw sequence data from the previously published studies were retrieved from the NCBI SRA Jones, et al. [29]; accession number PRJNA356981 and Nielsen, et al. [22]; accession number PRJNA396430) using the SRA Toolkit software.

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
