# Peer review of "Is the Intestinal Bacterial Community in the Australian Rabbitfish Siganus fuscescens Influenced by Seaweed Supplementation or Geography?"

_microorganisms, 2022, doi:10.3390/microorganisms10030497_

Round 1

Reviewer 1 Report

In the manuscript entitled: “Is the intestinal bacterial community in the Australian rabbitfish Siganus fuscescens influenced by seaweed supplementation or geography?” authors presents the results of influence of 15 diets on microbiome and microbiome-derived immunomodulatory potential in  Australian rabbitfish

Authors presented well designed study. Introduction is sufficient, methods are sound. Results are widely presented. Discussion is clear and conclusions are supported by results. The article is publication-worthy and interesting. Please find only minor comments/questions:

Minor issues:

  1. Line 32: including their metabolism
  2. Lines 78-79: Please change the reference (HU, et al., 2008; Li, et 78 al., 2018b; Osako, et al., 2006; Zhang, et al., 2016) to the number
  3. Same issue: line 89-90 (Trevathan- Thackett et al. 2019)
  4. Authors mentioned that newly captured fish were treated with hydrogen peroxide. Could authors explain how did it may influence the GI microbiome, which was actually investigated? In my opinion in conducting such experiments it would be better to proceed with all natural-existing parasites or pathogens, since in aqua-cultured fish they normally exist.
  5. Line 162: Authors stated that fish were fed one of 16 experimental diets while it was 15 diets used, does this number includes control diet?

Author Response

We wish to thank reviewer 1 for their constructive feedback. We have organised the comments (including those added directly to the manuscript) in the following table, together with our responses, for ease of review and highlighting the track-changes in the re-submission. We believe that the changes have improved the overall content and clarity of the manuscript. A clean copy of the manuscript together with a track-changes version has been uploaded for comparison.

Reviewer 2 Report

In this study, the author investigated the effects of those same seaweed species and four commercial feed supplements that used in aquaculture on the bacterial communities in Australian rabbitfish hindgut. The result showed a conserved core microbiome may exist within the hindgut of this fish species and provide other baseline data about temporal and spatial variation in the hindgut bacterial communities. However, the sample sizes are too small, for example the number of fish individuals in the treatments and the control group (n=3). The author should increase the sample size for analyzing the hindgut microbiome. Moreover, the arrangement of tanks for small and large fish should describe clearly in experiment design. The author mentioned three replicate tank per treatment in line 162-163; however, the total amount of fish has only 144 fish? In addition, the measurement of innate immunity is not suitable in 14days, normally the innate immune response appeared in the early days of treatment. The author should also check both innate and adaptive immune response by qRT-PCR. It is also not clear why the authors analyzed the innate immune responses of blood (line 192), if that was not the subject of their study, it makes difficult to understand the message the authors want to pass to the reader. The immune organ should also collect for the immune response analysis. Without any doubt, the intestinal microbiomes of all organisms with a digestive track respond to diet (quality and quantity). Besides age of the host, diet has to be assumed to be a highly important factor impacting the composition of intestinal microbiomes. In Figure 4, the large disparity in distributional pattern of individual difference. In summary, due to the lack of appropriate controls, it is not possible to conclude that the differences in microbiome composition were caused by the seaweed.

Minor comments:

  1. In line 32, “heir” should be “their”.
  2. In line 88-89, the sentences “[e.g., 50, 90 and 100% prevalence cut-offs; 23]” should be check.
  3. In line 89, “Trevathan Thackett et al. 2019” should cite as references.
  4. In line 90, the weight and length are average? The author should describe clearly.
  5. The author should explain the reason for analyzed hindgut microbiota in introduction.
  6. In line 284, “nMDS” should be “NMDS”.

Author Response

We wish to thank reviewer 2 for their constructive feedback. We have organised the comments (including those added directly to the manuscript) in the following table, together with our responses, for ease of review and highlighting the track-changes in the re-submission. We believe that the changes have improved the overall content and clarity of the manuscript. A clean copy of the manuscript together with a track-changes version has been uploaded for comparison.

Reviewer 3 Report

Review for: Is the intestinal bacterial community in the Australian rabbitfish Siganus fuscescens influenced by seaweed supplementation or geography?

By Thépot et al.

The authors present an interesting study on the microbial composition of  hindguts in rabbitfish. The authors test two clearly formulated hypotheses regarding the potential effects of supplemental feeding and geography on intestinal microbial composition in this species. This knowledge is important because this and other fish species are increasingly bred in fish farms to ensure food security. Intestinal microbiota composition has been linked to immunological responses and therefore health conditions. Hence, this study is timely and will help to inform management and industry. The manuscript is well-written and therefore, I have only minor corrections as outlined below.

Since the study deals with different populations across Australia, I recommend adding a map with the sampling locations so that people outside of Australia understand better where these locations are.

Page 2, line 63: It looks like there is an extra space between ‘seaweed’ and ‘caused’. Please check and revise.

Page 2, line 68: ‘microbial mediated’ should be ‘microbially-mediated’ as in sentences further down.

Page 2, line 78: ‘HU’ should be ‘Hu’.

Page 5, line 246: ‘bellow’ should be ‘below’.

Page 7, line 329: Remove extra space between ‘had’ and ‘significantly’.

Page 9, Fig. 3: It seems that this figure has lost resolution due to conversion, maybe. Please make sure that the figure quality is high enough in the final publication as some of the numbers are a bit hard to read.

Page 11, line 414: I think, the ‘the’ in this text fragment can be deleted: which was the lower in fish fed control diets compared to

Page 12, line 435: Change ‘6290’ to ‘6,290’ for consistency.

Page 16, lines 596 and 598, 599, 608: The species names mentioned here should be written in italic font. Also, in line 599: Add semicolon and space between ‘Siganus canaliculatus53,’.

Page 16, line 618: delete extra spacing after ‘Ours’.

Page 17, line 679: Change ‘4000’ to ‘4,000’ for consistency.

Author Response

We wish to thank reviewer 3 for their constructive feedback. We have organised the comments (including those added directly to the manuscript) in the following table, together with our responses, for ease of review and highlighting the track-changes in the re-submission. We believe that the changes have improved the overall content and clarity of the manuscript. A clean copy of the manuscript together with a track-changes version has been uploaded for comparison.

Round 2

Reviewer 2 Report

The author revised all my comments.